# Damping Capacity and Storage Modulus of SiC Matrix Composites Infiltrated by AlSi Alloy

**Xuan Li [1], Yongzhe Fan [1], Xue Zhao [1], Ruina Ma [1,\*], An Du [1,\*], Xiaoming Cao [1,2] and Huiyun Ban [1]**

1   Key Lab for New Type of Functional Materials in Hebei Province, Tianjin Key Lab Material Laminating Fabrication and Interface, School of Material Science and Engineering, Hebei University of Technology, Tianjin 300132, China; nblixuan@126.com (X.L.); fyz@hebut.edu.cn (Y.F.); zhaoxue@hebut.edu.cn (X.Z.); gd_sam@galvanize.com.cn (X.C.); banhuiyun@126.com (H.B.)

2   Tianjin Gongda Galvanizing Equipment Co., Ltd., Tianjin 300132, China

\*   Correspondence: maryna@126.com (R.M.); duan@hebut.edu.cn (A.D.);
    Tel.: +86-136-4207-4657 (R.M.); +86-136-0209-7216 (A.D.)

**Abstract:** In this paper, we describe how an aluminum alloy-reinforced silicon carbide ceramic matrix composite (SiCCMC) with excellent damping capacity and storage modulus was fabricated by infiltration. The effects of silicon (Si) on the microstructure and damping capacity of the composite were studied. The interface bonding and damping mechanism involved were also discussed. The results show that composites with high damping capacity can be obtained by infiltrating SiC ceramics with aluminum alloy. The residual Si in the SiC ceramic had little effect on the damping capacity, and it provided the passage of aluminum alloy into the interior of the SiC ceramic. The aluminum atoms penetrate the SiC particles by diffusion. Optimal composite damping capacity was obtained when the Si content in the aluminum alloy was 15 wt. %, because the AlSi/SiC interface friction dissipated most of thermal energy. $Ti_3SiC_2$ formed on the surface had little effect on the damping capacity. Additionally, by changing the Si content in the aluminum alloy, the strength and damping capacity of the composites can be controlled.

**Keywords:** SiC ceramic matrix composite; AlSi alloy; damping capacity; storage modulus

## 1. Introduction

Silicon carbideceramic matrix composite (SiCCMC) is known for their high-temperature strength and excellent wear resistance and are being applied in many fields, especially in branches of structural engineering applications (such as satellite attitude control, cruise missile, and automotive engine), where substantial dynamic vibrations occurring during normal operation, cause a reduction in the system accuracy and service time of the structural members. The vibration and resulting noise of SiC, which are due to the temperature-independent low intrinsic damping of SiC [1], are important issues to be tackled [2–6]. To solve these problems, it is necessary to develop composites with high damping properties [7–9]. Cheng [10,11] reported the damping behavior of carbon fiber-reinforced SiC (C/SiC) composites. Presently, the research on SiCCMC focuses mainly on conventional properties and manufacturing processes, and research on its damping properties is lacking. Aluminum (Al) and its alloys are widely used, due to their low density and high strength [12–15]. It has been reported that adding SiC particles to the Al matrix improves the damping capacity of composites [7,16,17]. Therefore, we sought to develop an Al alloy-reinforced SiCCMC material with excellent high-temperature mechanical properties and damping properties.

Due to the inherent brittleness of SiC ceramic and the relatively poor wettability between ceramics and Al melt [18], the interfacial bonding strength of Al/SiC composites is not high. A brittle

aluminum carbide ($Al_4C_3$), phase can easily be formed at the interface between SiC and Al at high temperatures [19,20]. Since this phase is highly sensitive to moisture [21,22], it results in poor interface bonding and strength. Unsatisfactory interface bonding makes it difficult for composite materials to achieve the desired strength performance index. Zhang et al. [23] reported that, adding appropriate quantities of silicon (Si) and titanium (Ti) into Al alloy is beneficial to decrease the wetting angle of molten metal to SiC ceramics and to improve the strength of interface bonding. Furthermore, it was reported that the reaction between the Al melt and SiC was inhibited when an appropriate quantity of Si was added to the Al alloy [24]. This paper is concerned with the influence of Si content in Al alloy on the storage modulus, damping properties, and interfacial bonding of AlSi/SiC composites, produced by infiltration.

## 2. Materials and Methods

### 2.1. Fabrication of AlSi/SiC Composites

Composites with different Si content were fabricated by infiltration. The SiC ceramic was produced by Huamei Ceramics Co., Ltd., Shandong, China, and consisted of two phases, SiC and Si. The volume fraction of Si is 16% in the SiC ceramic (Figure 1). The Al-10 wt. % Ti alloy was produced by Nonferrous Metals Co., Ltd., Baoji, China. The Al-10 wt. % Ti alloy was placed inside a graphite crucible and heated to 900 °C in an electric furnace to melt it, the 5, 10, 15, and 20 wt. % polycrystalline Si wafers were put into the molten alloy, and heated to 900 °C for 1 h. SiC was polished with a 1000-grit diamond grinding disc and cleaned in alcohol for 20 min. Subsequently, the SiC ceramic was put into the molten alloy and maintained at 900 °C for 6 h.

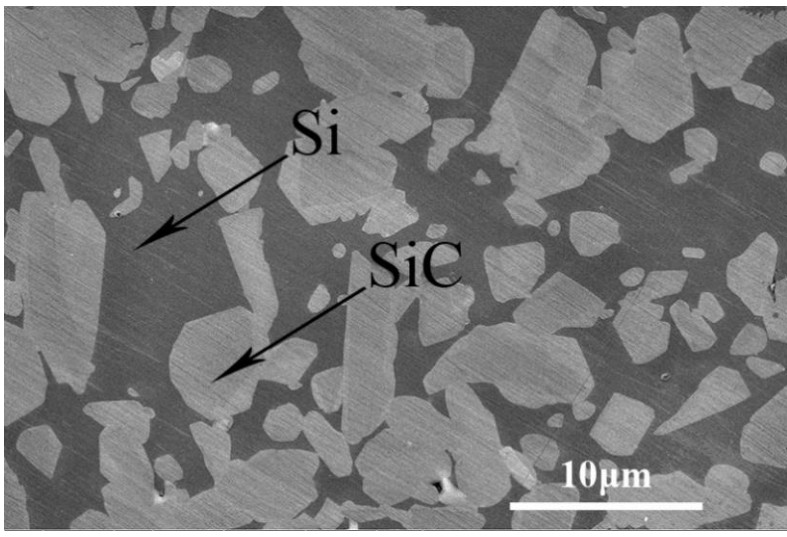

**Figure 1.** SEM (scanning electron microscopy) image of the initial SiC ceramic.

### 2.2. Microstructure and Phase Composition

Microstructural analysis was performed using scanning electron microscopy (SEM, Nova Nano SEM 450 FEG, FEI, Hillsboro, OR, USA), and the chemical composition was analyzed using energy-dispersive X-ray spectroscopy (EDS, EDAX, FEI, Hillsboro, OR, USA). The samples phases were characterized by X-ray diffraction (XRD, D8 Advance, Bruker, Karlsruhe, Germany) using Cu Kα radiation. The elemental species and chemical valence of the sample surface were analyzed by X-ray photoelectron spectroscopy (XPS, ESCALAB 250Xi, Thermo Fisher, Waltham, MA, USA). Raman spectroscopy (InVia Reflex, Renishaw, Wotton-under-Edge, UK) was utilized to analyze the SiC crystal structure of a selected sample microdomain, using the 532 nm line of an Ar ion laser as the excitation source.

*2.3. Dynamic Mechanical Analysis (DMA)*

Wire-electrode cutting was used to obtain specimens of dimensions of 60 mm × 10 mm × 2.5 mm for the DMA test. The test was performed in a dynamic mechanical analyzer (DMA, Q800, TA Instruments, New Castle, UK) in the dual cantilever testing mode. DMA tests were performed at three frequencies, 0.5, 1, and 5 Hz, by using a maximum dynamic force of 18 N for a maximum amplitude of 1.5 μm. Measurements were acquired from 40 °C to 350 °C, during the heating phase, at a rate of 3 K/min. In this study, the damping capacity was evaluated by analyzing the loss factor, tan delta, using the relation:

$$\tan \delta = \frac{E''}{E'} \tag{1}$$

where E″ is the loss modulus, corresponding to the viscous component. The storage modulus, E′, relates to the stiffness of the material.

## 3. Results and Discussion

The results of microstructure and chemical composition analyses are discussed in order to investigate the AlSi/SiC interfacial reaction. The storage modulus and damping capacity of composites are subsequently presented and discussed. The influence of the AlSi phase and interface on these properties is studied. Finally, the damping mechanism involved is discussed.

*3.1. Microstructure*

Figure 2a–d shows the cross-sectional microstructure of SiC ceramics infiltrated by Al alloy with 5, 10, 15, and 20 wt. % Si, respectively. As shown in Figure 2a–c, a white reaction layer was detected on the surface of the samples after infiltration for 6 h. On the contrary, there were no obvious reaction layers on the surface of the SiC ceramic after infiltration by Al alloy with 20 wt. % Si (Figure 2d).

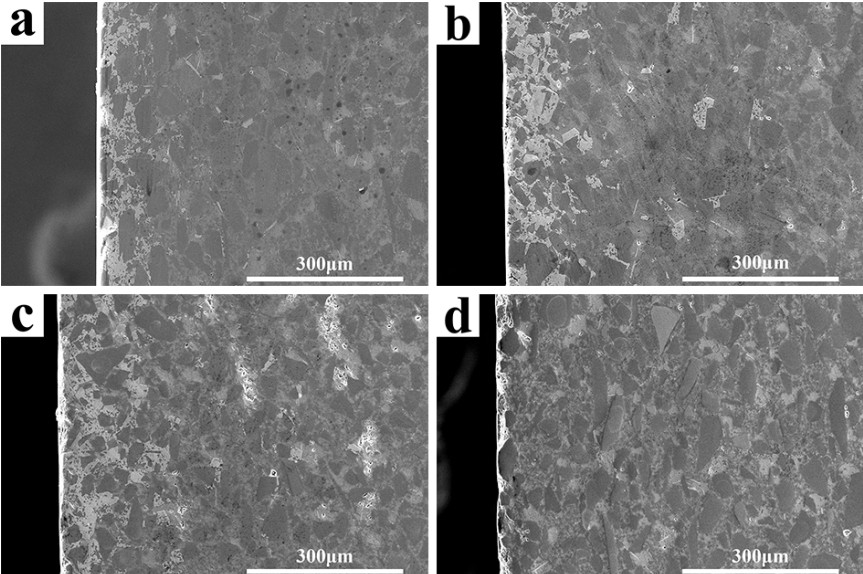

**Figure 2.** Microstructure of SiC ceramic infiltrated by Al alloy with different Si content: (**a**) 5 wt. %, (**b**) 10 wt. %, (**c**) 15 wt. %, and (**d**) 20 wt. %.

As shown in Figure 3, the XRD patterns of the surfaces of the SiC ceramic infiltrated by Al alloys with different Si content indicate that SiC and $Ti_3SiC_2$ are the main components of the surface of SiC when the Si content is less than 20 wt. %. The diffraction peaks of $Ti_3SiC_2$ increases first and then decreases with increasing Si content. In contrast, the samples infiltrated by Al alloy with 20 wt. % Si mainly contain SiC, Al, and Si. The XRD and micrographs results show that the white reaction phase is

Ti$_3$SiC$_2$. Ti$_3$SiC$_2$ has a thickness of about 150 μm. The volume fraction (vol.%) of each phase of the composite surface is shown in Table 1.

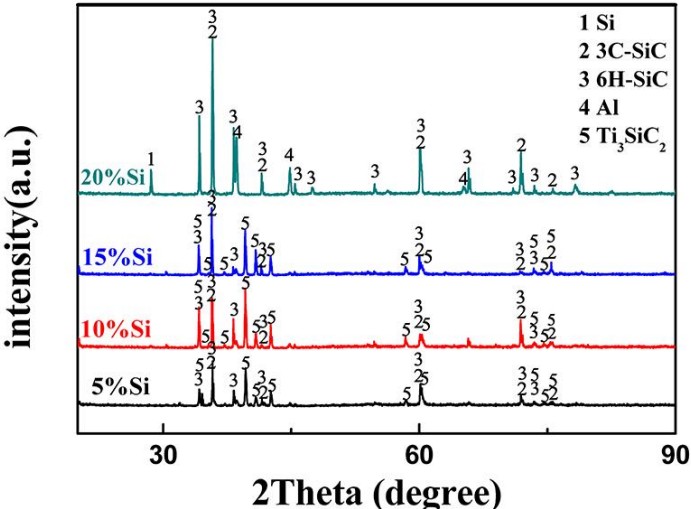

**Figure 3.** X-ray diffraction (XRD) pattern of the surfaces of the SiC ceramic.

**Table 1.** Volume fraction (vol.%) of surface phase.

| Si Content | 6H-SiC | 3C-SiC | Ti$_3$SiC$_2$ | Al | Si |
|---|---|---|---|---|---|
| 5 wt. % Si | 65 | 23.8 | 11.2 | - | - |
| 10 wt. % Si | 14.3 | 3.3 | 82.4 | - | - |
| 15 wt. % Si | 72.4 | 17.2 | 10.4 | - | - |
| 20 wt. % Si | 73 | 16.6 | - | 8.4 | 2 |

In order to study the infiltration process of Al-based alloys in SiC materials, the body center of the sample was observed. The microstructure of the sample center is shown in Figure 4. The amount of Si in the four different composites was 5, 10, 15, and 20 wt. %, respectively. A transition layer between SiC and AlSi was detected. EDS results (Table 2) show that the Si phase in the initial ceramic becomes AlSi after infiltration. This indicates that Al alloy is dissolved and infiltrated along the position of Si in the ceramic SiC. When Al alloy contact with Si in SiC ceramic for a certain time at 900 °C, AlSi eutectic with a low melting point will be generated to provide an alloy diffusion channel. Furthermore, Al is detected in the transition layer between SiC and AlSi, and a core of small particle size SiC. The average atomic content of Al in SiC first increases and then decreases. When SiC is infiltrated by Al alloy with 15 wt. % Si, the SiC has the highest atomic atom content of Al (15.24 at. %).

To further observe the microstructure at the interface of the composites, the results of XRD, XPS, and Raman analysis of the interface are provided in Figures 5 and 6. The XRD patterns (Figure 5a) show that no interfacial reaction was found in the core of the composites. The volume fraction (vol.%) of each phase of the center is shown in Table 3. A change in the chemical composition in the vicinity of the boundary between the SiC and AlSi is shown in Figure 6. It is clear that Al from the AlSi alloy enters the SiC structure, and it can be inferred that the Al atoms diffuse between smaller SiC particles. In the XPS spectrum of the interface, shown in Figure 5b, two peaks of the Al 2p can be observed, which are assigned to AlO bond and the AlAl bond. No Al$_4$C$_3$ produced by SiC/Al reaction was clearly observed; if present, peak would be observed at 73.6 eV. Al has low bonding energy at the Si site on the SiC lattice [25] and substitutional diffusion can occur by replacement of Si with Al. Figure 5c shows the Raman spectrum of the interface between AlSi alloy and SiC. The Raman spectrum shows the presence of sharp peaks at 788 and 966 cm$^{-1}$, which correspond to the transverse optical (TO) phonon mode and the longitudinal optical (LO) phonon mode peaks of SiC, respectively [26]. The movement of Raman

spectral peaks is closely related to the crystal structure of the material. An AlSiC solid solution was formed by substituting Al atoms instead of Si atoms in the SiC lattice, which led to an increase in the hole concentration of the crystal. This weakens the peak intensity of 966 cm$^{-1}$, and the peak position shifts to a higher wavenumber. Arsenault et al. [27,28] confirmed that Al in the composite diffused into SiC through the interface, and the diffusion distance was higher than the calculated value, but there was no diffusion of silicon (Si) and carbon (C) into the aluminum (Al) liquid. These studies all suggest that the diffusion bonding of SiC with Al results in a high bond strength of the matrix/enhancement phase. Therefore, it can be speculated that the transition layer between AlSi/SiC is a diffusion layer, which improves the interface bonding strength. The interface bonding strength changes with the content of Si in the infiltrated Al alloy.

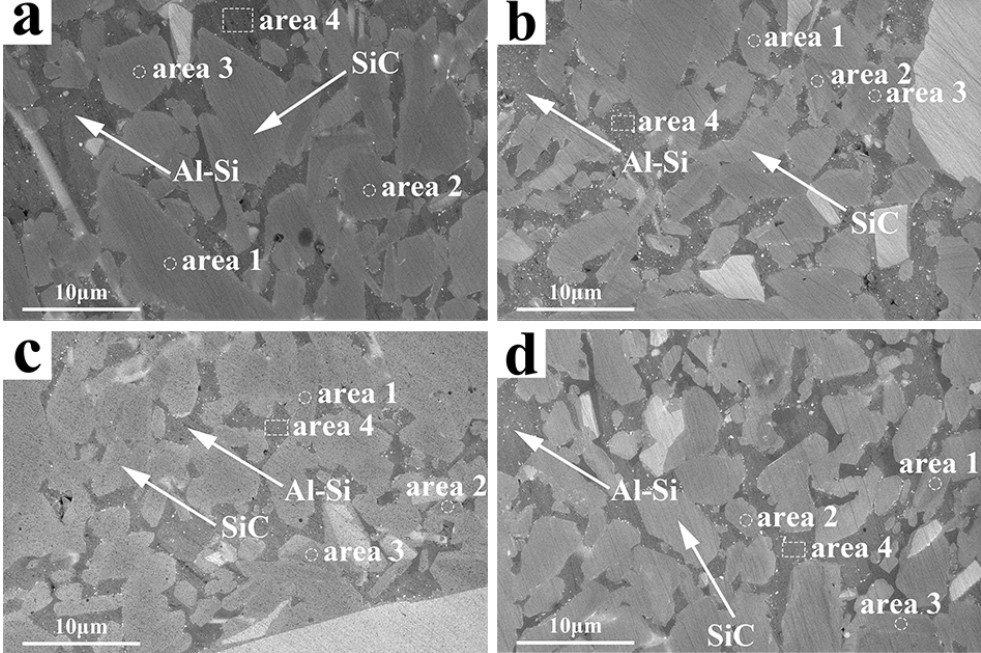

**Figure 4.** SEM image of the SiC ceramic infiltrated by Al alloy with different Si content: (**a**) 5 wt. %, (**b**) 10 wt. %, (**c**) 15 wt. %, and (**d**) 20 wt. %.

**Table 2.** Chemical composition of marked areas on Figure 4, obtained by energy-dispersive X-ray spectroscopy (EDS).

| Area | 5 wt. % Si | 10 wt. % Si | 15 wt. % Si | 20 wt. % Si |
|---|---|---|---|---|
| **area1** | | | | |
| **Al** | 3.01 | 8.78 | 13.84 | 10.51 |
| **Si** | 37.87 | 44.3 | 47.62 | 38.63 |
| **C** | 59.13 | 46.92 | 38.54 | 50.85 |
| **area2** | | | | |
| **Al** | 2.38 | 7.6 | 17.52 | 9.32 |
| **Si** | 47.39 | 16.36 | 47.26 | 47.64 |
| **C** | 50.23 | 46.05 | 35.22 | 43.04 |
| **area3** | | | | |
| **Al** | 2.67 | 9.08 | 14.38 | 8.47 |
| **Si** | 65.49 | 45.94 | 50.11 | 47.37 |
| **C** | 31.84 | 44.98 | 35.51 | 44.15 |
| **average Al content** | 2.68 | 8.48 | 15.24 | 9.43 |
| **area4** | | | | |
| **Al** | 28.37 | 56.39 | 28.55 | 71.81 |
| **Si** | 49.56 | 32.36 | 56.49 | 15.9 |
| **C** | 22.02 | 11.25 | 14.96 | 12.29 |

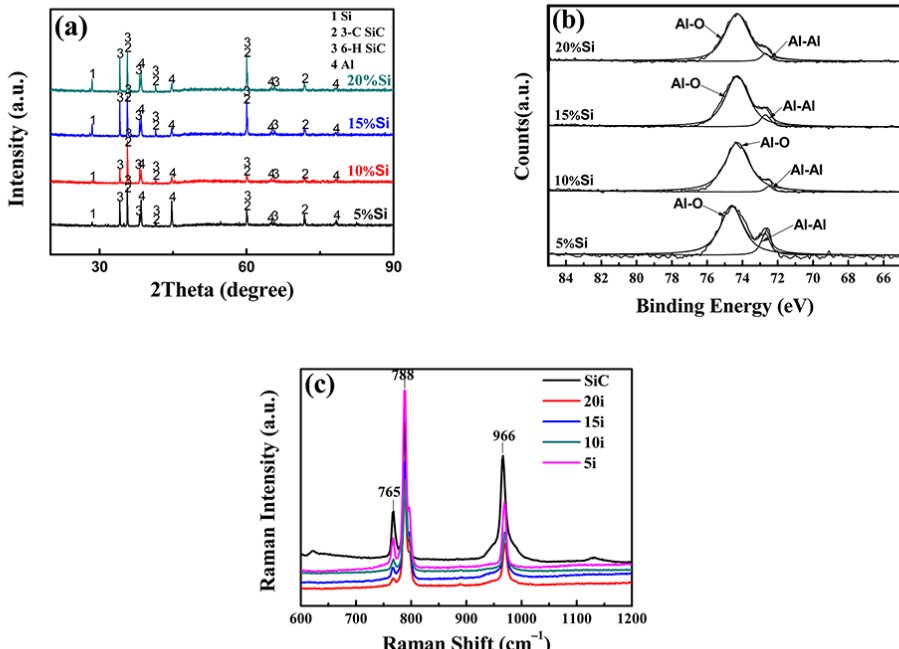

**Figure 5.** XRD patterns (**a**), X-ray photoelectron spectroscopy (XPS) spectra (**b**) and Raman spectra (**c**) of the composites.

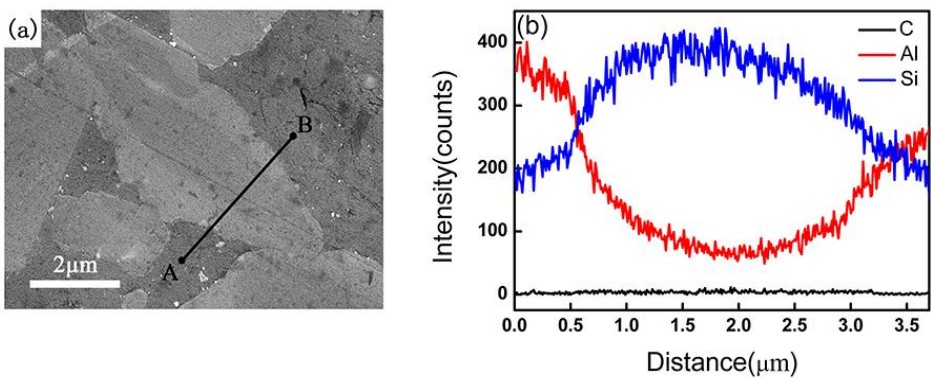

**Figure 6.** Line sketch of SiC with transition layer from point A to point B (**a**). EDS line scanning analysis of the transition layer (**b**).

**Table 3.** Volume fraction (vol.%) of central phase.

| Si Content | 6H-SiC | 3C-SiC | Al | Si |
|---|---|---|---|---|
| 5 wt. % Si | 65.9 | 22.5 | 10.9 | 0.7 |
| 10 wt. % Si | 73.4 | 18.4 | 7.7 | 0.5 |
| 15 wt. % Si | 74.1 | 19.9 | 5.5 | 0.5 |
| 20 wt. % Si | 68.6 | 16.9 | 9.8 | 4.7 |

### 3.2. Damping Behavior

Figure 7 shows the storage modulus of SiC and the composites. It can be observed that the test frequencies of 0.5, 1 and 5 Hz had little effect on the storage modulus. For all test frequencies (0.5, 1, and 5 Hz), it is clear that the storage modulus of SiC without infiltration was higher than that of the composites in the range from 40 °C to 350 °C. The composites infiltrated by Al alloy with 15 wt. % Si had the highest storage modulus (210 GPa) at 40 °C, and the composites infiltrated by Al alloy with 10 wt. % Si had the highest storage modulus (155 GPa) at 350 °C. The implication is that the

energy absorption capacity of the composites increases first, and then decreases with increasing Si content in Al alloy. Wang [29] reported that the bending strength of the composite was highest when the Si content in the Al alloy was 10 wt. %. This is because the AlSi alloy composition is close to the eutectic point, and massive primary Si did not exist during cooling and solidification of the AlSi phase. Furthermore, the secondary Si precipitates dispersed in the AlSi phase increase the strength of the material. There is a proportional relationship between the bending strength and the elastic modulus, and an increase in the bending strength of the material means an increase in the elastic modulus, which confirms that the composites infiltrated by Al alloy with 10 and 15 wt. % Si have the highest storage modulus. It can also be observed that the storage modulus decreases with an increase in temperature, because Al in composites becomes soft with increasing temperature, which reduced the stiffness of composites. The decrease in the dynamic stiffness of the composites with temperature contributes to the storage modulus, which decreases with increasing of temperature [30], as composites maintained high strength over the temperature range of 40–350 °C.

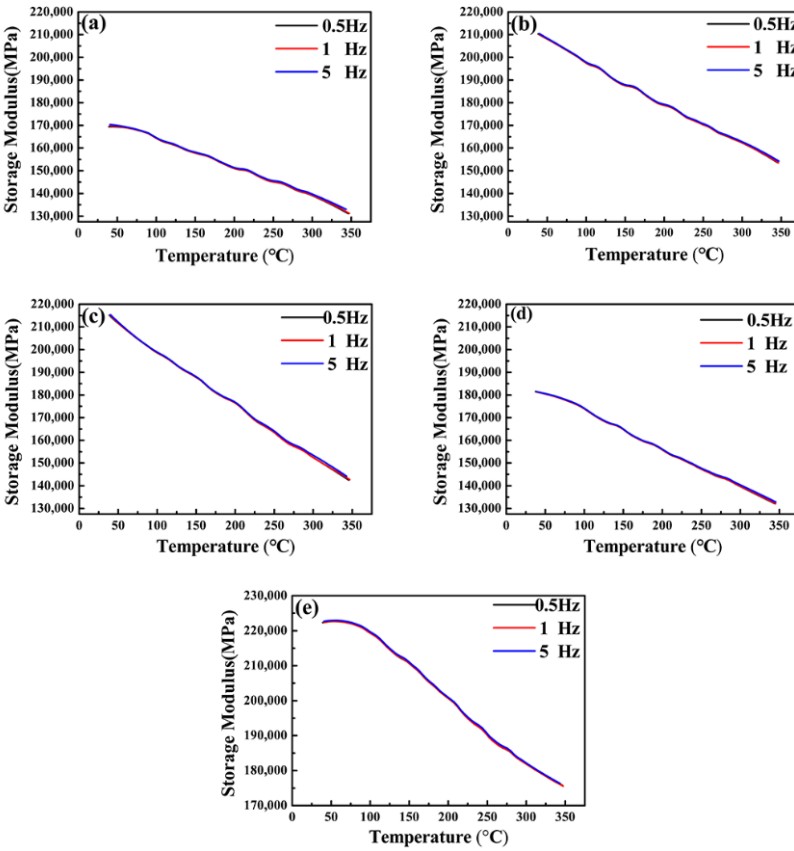

**Figure 7.** Storage modulus of the SiC ceramic infiltrated by Al alloy with different Si content: (**a**) 5 wt. %, (**b**) 10 wt. %, (**c**) 15 wt. %, (**d**) 20 wt. % and (**e**) initial SiC ceramic.

The damping capacities of the SiC and the composites infiltrated by Al alloy with 5, 10, 15, and 20 wt. % Si are presented in Figure 8. Similar trends in damping capacity change with temperature were found in different composites at all test frequencies. It is clear that the damping capacities of all the composites tested at 5 and 1 Hz were lower than those at 0.5 Hz on the temperature range of 40–350 °C. The damping capacity of all composites was observed to increase with temperature in the range of 40–125 °C, remained constant at 125–250 °C, and increased in the 250–350 °C. Two damping peaks at 100–150 °C and 325–350 °C were observed, denoted as P1 and P2, respectively. The high damping capacity of the composites is attributed to two principal factors: 1) the intrinsic damping properties of the Al alloy and SiC matrix; 2) the combination status of the interface between the AlSi alloy and SiC. The intrinsic damping capacities of Al and SiC are $3.6 \times 10^{-3}$ and $4 \times 10^{-4}$, respectively,

at room temperature and low frequency [7]. By predicting the damping capacity of composites by the rule of mixtures (ROM), it can be inferred that the damping capacity of composites is less than 0.0036, which means that the intrinsic contribution is not significant. Silva Prasad [31] reported that high interfacial strains and stresses are generated during the cooling and solidification of composites when there is a great difference in the coefficient of thermal expansion (CTE) between the reinforcement and the matrix, which leads to a degree of dislocation that contributes to the improvement in damping capacity. The coefficient of thermal expansion of the AlSi reinforcements ($17.8{\sim}24 \times 10^{-6}$ K$^{-1}$) and the SiC matrix (~$4 \times 10^{-6}$ K$^{-1}$) [32,33] were significantly different. The mechanism of damping through dislocations from room temperature to 125 °C and low frequency can be explained by the Granato-Lucke (G-L) mechanism [34]. Dislocations near the interface were pinned by some weak pinning sites (solute atoms and vacancies), which dissipate energy by reciprocating motion between the weak pinning points such as elastic string under cyclic loading. Higher damping capacity at low frequency was a result of the fact that reciprocating motion of dislocations swept a larger area and dissipated more energy. The damping mechanism of the P1 peak conformed to the dislocation-induced mechanism. After this point, the dislocation density decreased with increasing temperature, which means that the contribution of the dislocation-induced damping mechanism to the damping can be neglected. At higher temperatures, the interfacial bonding strength between SiC and AlSi phase was weakened because the Al became less stiff, which increased the interface slip distance. P2 was believed to be formed by sliding friction and had thermal activation. The damping capacity increased due to thermal energy dissipation. It was reported that the wettability of Al alloy and SiC at 900 °C increased first and then decreased with increasing Si content in the alloy [35,36]. When the Si content was 12 wt. %, the wetting angle was the lowest, and the interface bonding strength was the highest. If the interface bonding is too weak, although the slip distance of the interface increases, the interface friction energy will not increase constantly due to the weak interface bonding. Therefore, by controlling the infiltration of Al alloys with different Si contents to form an optimal interface, the maximum energy dissipation can be produced, and the damping properties of the composite material can be improved. There was Ti$_3$SiC$_2$ layer on the surface of the composites infiltrated by Al alloy with 5 wt. % Si. The surface of the composites infiltrated by Al alloy with 20 wt. % Si had no Ti$_3$SiC$_2$ layer. Damping capacity was not significantly different between the above composites. As the phase formed on the surface, Ti$_3$SiC$_2$ mainly improved the wear resistance of the composite and had little effect on the damping capacity. From the results, we can see that the composite having the best high temperature damping capacity is the one infiltrated by Al alloy with 15 wt. % Si, which is one time higher than the SiC without infiltration, meaning that the sliding friction of the interface has dissipated the maximum energy, and the interface combination is optimal for damping.

### 3.3. Influence of Si

In the process of sintering, it is difficult to obtain dense SiC ceramic due to the low diffusion coefficient, but SiC ceramic with high density can be obtained by reaction-sintering. However, the residual Si produced by the reaction-sintering reduces the strength of the ceramic [37,38]. When SiC ceramic infiltrated by the Al10TixSi alloy, the aluminum alloy melts the residual Si of the SiC ceramic, and the position of the residual Si provides the passage of the aluminum alloy into the center of the ceramic. According to the AlSi phase diagram, the eutectic alloy Al-12.5 wt. % Si has the lowest melting point and good fluidity. Under this condition, the aluminum alloy infiltrates the center of the composite and full contact with SiC, and the interface bonding strength is high. In this experiment, the 10, 15 wt. % Si content of the aluminum alloy is closest to the eutectic alloy composition, so the storage modulus is higher. Since the residual Si is replaced by the Al alloy, the hardness of the composite decreases, resulting in the storage modulus of the composite lower than the SiC ceramic. Si has a high melting point and is less affected by temperature. At the temperature of 300–350 °C, the interface of SiC/Si does not generate sliding friction and has no thermal energy dissipation, so the damping capacity of SiC ceramics do not change at higher temperatures.

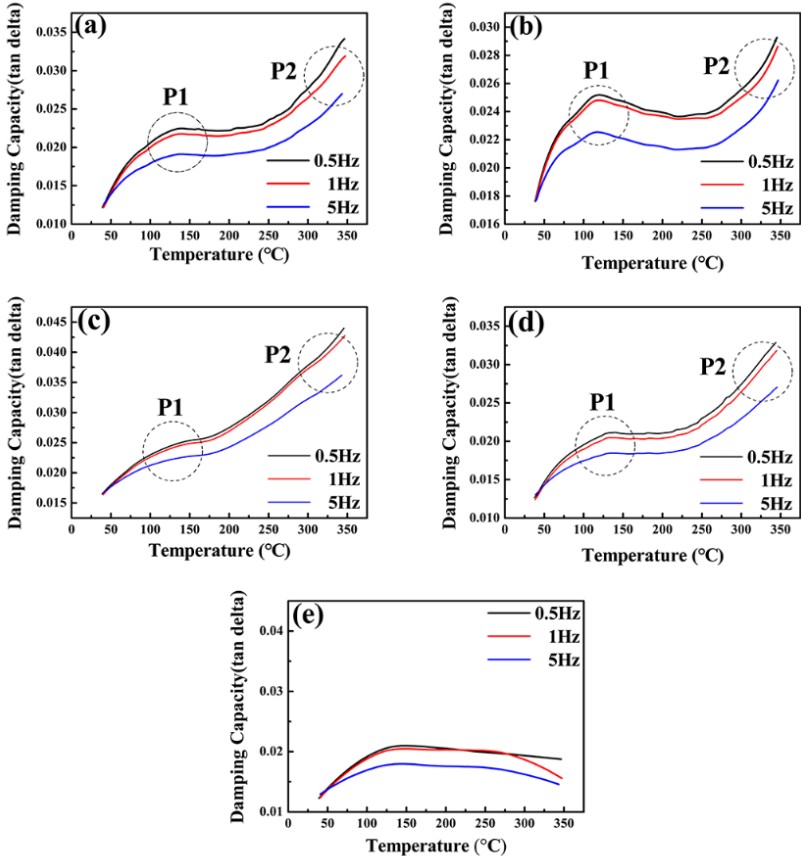

**Figure 8.** Damping capacity of the SiC ceramic infiltrated by Al alloy with different Si content: (**a**) 5 wt. %, (**b**) 10 wt. %, (**c**) 15 w. t%, (**d**) 20 wt. %, and (**e**) initial SiC ceramic.

## 4. Conclusions

- All SiCCMCs were successfully fabricated by infiltration.
- Different interface combination statuses were obtained by infiltrating Al alloys with 5, 10, 15, and 20 wt. % Si, A white reaction layer was formed on the surface of the composites when the Si content is less than 20 wt. %. No interfacial reaction was detected in the core of all composites. The Al alloy was combined with SiC through diffusion.
- The storage modulus of the composites reached 210 GPa or more at room temperature and 140 GPa or more at 350 °C when the Si content was 10 and 15 wt. %, respectively. The reason for this is twofold: the brittle primary Si was not precipitated during the cooling process of the composites, and the dispersed secondary Si increased the strength of the composites.
- The composites exhibited the best damping capacity at high temperature when the Si content was 15 wt. %, because the interface friction dissipated most of thermal energy.
- The strength and damping capacity of the composites can be controlled by changing the Si content in the Al alloy to facilitate different applications.

**Author Contributions:** Writing-original draft, X.L.; Investigation, X.L; Resources, Y.F. and X.Z.; Project administration, R.M.; Writing-review and editing, A.D.; Supervision, X.C.; Validation, H.B.

**Funding:** This research was funded by the National Natural Science Foundation of China (grant number 51601056), the Natural Science Foundation of Hebei Province of China (grant number E2017202012) and Innovation and the Entrepreneurship Training Program for College Students of Hebei University of Technology (grant number X201910080058).

**Conflicts of Interest:** The authors declare no conflict of interest.

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
