# Peer review of "Damping Capacity and Storage Modulus of SiC Matrix Composites Infiltrated by AlSi Alloy"

_metals, doi:10.3390/met9111195_

Round 1

Reviewer 1 Report

The following points must be specified and clarified before acceptation of the acceptation of the manuscript:

The starting composition of material must be indicated: Al, Si and SiC proportions The nature of phases in both XRD figures must be harmonized. For instance the nature of SiC (3-C and 6-H) must be indicated in Fig 3 similarly to Fig.5 Quantification of the phases present in the samples will help to better understand the observed properties. This quantification can be obtained by XRD analysis by Match program for instance. XRD results seem different at the surface and at the central part. Authors must define more precisely this central part. Does it mean the body center of the sample? wt% 15 Si seems the better composition . The properties were tentatively explained on the basis of Al, Si and SiC presence. However the sample contains also Ti3SiC2 which has excellent thermal and mechanical properties (great number of papers is published). What is the role of this component on the observed properties?

Reviewer 2 Report

The submitted work titled "Damping capacity and storage modulus of SiC matrix composites infiltrated by Al-Si alloy" is interesting, but, the authors need to carefully consider the reviewers' comments to further improve the manuscript for further consideration.

Under abstract, the words "high temperature mechanical properties" has to be removed as other than the modulus and damping capacity (at high temperatures), no other mechanical properties are discussed in the paper. Under section 2.1, fabrication of Al-Si/SiC composites, line 53, is the material Al-10wt.%Ti? Authors need to carefully check the materials whether Al-Si? Table 1, EDS is semi-quantitative. The authors are requested to add EDS elemental maps for the distribution of Al and Si/SiC.  Under the results and discussion, apart from the Si present with the Al-Si alloy, the role of residual Si (16 vol.%, line 53) within the SiC has to be discussed. The results indicate Si presence improve the damping capacity and modulus properties. Otherwise, the results are misleading since the authors neglect the Si within the SiC ceramics. Authors may need to add a paragraph under results and discussion section and clearly explain the role of Si (both from Al alloy and SiC ceramics).  

Round 2

Reviewer 1 Report

Authors have taken into account the major remarks and suggestions. The paper can now be accepted.

Reviewer 2 Report

The authors have carefully addressed the reviewers' comments and the revised version can be considered for publication.